# Baseline Characteristics of Weight-Loss Success in a Personalized Nutrition Intervention: A Secondary Analysis

**DOI:** 10.3390/nu17132178

**Published:** 2025-06-30

**Authors:** Collin J. Popp, Chan Wang, Lauren Berube, Margaret Curran, Lu Hu, Mary Lou Pompeii, Souptik Barua, Huilin Li, David E. St-Jules, Antoinette Schoenthaler, Eran Segal, Michael Bergman, Mary Ann Sevick

**Affiliations:** 1Department of Population Health, New York University Langone Health, 180 Madison Ave, New York, NY 10016, USA; 2Public Health Nutrition Program, Department of Epidemiology, NYU School of Global Public Health, 708 Broadway, New York, NY 10003, USA; 3Department of Medicine, New York University Langone Health, New York, NY 10016, USA; 4Department of Nutrition, University of Nevada, Reno, 1664 N. Virginia Street, Reno, NV 89557, USA; 5Department of Computer Science and Applied Mathematics, Weizmann Institute of Science, Rehovot 7610001, Israel

**Keywords:** precision nutrition, continuous glucose monitors, prediabetes, weight management

## Abstract

**Background/Objectives**: The aim of this secondary analysis is to determine the baseline characteristics that are associated with a higher likelihood of weight-loss success in a personalized nutrition intervention. **Methods**: Data were analyzed in adults with abnormal glucose metabolism and obesity from a 6-month behavioral counseling randomized clinical trial. Participants were randomized to two calorie-restricted diets: a low-fat diet (*Standardized*) or a personalized nutrition diet leveraging a machine learning algorithm (*Personalized*). The gradient boosting machine method was used to determine the baseline variables (i.e., age, weight-loss self-efficacy) that predicted successful weight loss (≥5%) at 6 months in each study arm separately, using repeated five-fold cross-validation with 100 repetitions. **Results**: A total of 155 participants (*Personalized*: n = 84 vs. *Standardized*: n = 71) contributed data (mean [standard deviation]: age, 59 [10] y; 66.5% female; 56.1% White; body mass index (BMI), 33.4 [4.6] kg/m^2^). In both arms, higher baseline self-efficacy for weight loss was a predictor of weight-loss success. Participants with a higher BMI (*p* < 0.0001) in the *Standardized* arm and those who were older (*p* < 0.0001) in the *Personalized* arm were more likely to achieve successful weight loss. **Conclusions**: Future weight-loss interventions may consider providing tailored behavioral support for individuals based on weight-loss self-efficacy, BMI, and age.

## 1. Introduction

The emerging field of precision nutrition focuses on understanding metabolic variability within and between individuals to develop personalized nutrition recommendations [1]. Providing more specific or tailored healthy eating guidelines has been hypothesized to address the inter-individual variability in response to dietary interventions and prevent the progression of chronic diseases such as obesity and type 2 diabetes (T2D).

Despite underlying inter-individual differences in the metabolic response to foods, recent findings have demonstrated null findings in weight loss between personalized nutrition interventions informed by genetic, microbiomic, and metabolomic data and control diets [2,3,4,5,6]. In a 12-week trial, Höchsmann et al. showed no significant difference in weight loss between genotype-concordant and -discordant diets among fat and carbohydrate responders, indicating that personalized diets based on genotype do not provide additional weight-loss benefits for individuals with overweight or obesity [6]. Our group found no difference in weight loss at 6 months between a low-fat diet (*Standardized*) and a personalized diet (*Personalized*) that leveraged an evidence-based machine learning algorithm to reduce the post-prandial glycemic response (PPGR) to foods [3]. However, there was considerable variability in the weight-loss response, with some individuals losing a clinically significant (≥5%) amount of body weight in both arms. A prior personalized nutrition intervention suggests there are characteristics of individuals who may benefit more from a personalized dietary intervention [7]. The Food4Me Study was a large, four-arm, randomized controlled trial conducted in seven European countries designed to examine the effects of personalized dietary and physical activity recommendations on the Healthy Eating Index at 6 months [5]. Livingstone et al. found that those benefiting most from a personalized nutrition intervention were older adults, women, and participants with poor diet quality at baseline [7]. Therefore, understanding how baseline characteristics predict outcomes in precision nutrition interventions may help researchers stratify large samples into unique subgroups to provide early, tailored support to participants and improve intervention effectiveness. Therefore, the objective of this study was to identify which baseline characteristics predicted weight-loss success (≥5%) at 6 months using a machine learning method.

## 2. Materials and Methods

### 2.1. Study Objective and Approvals

This study is a secondary analysis of data collected from The Personal Diet study, a randomized controlled trial comparing the effects of *Personalized* and *Standardized* diet interventions on weight loss in participants with prediabetes or moderately controlled T2D (ClinicalTrials.gov Identifier: NCT03336411). A detailed study design protocol and papers on the primary and secondary outcomes have been previously published [3,8,9]. The trial was conducted in accordance with the Declaration of Helsinki and was approved by the NYU Langone Health (NYULH) Institutional Review Board (IRB #17-00741). All study participants signed an informed consent prior to their participation. Data included in this analysis were collected at baseline (0) and 6 months.

### 2.2. Participants

Participants were recruited between January 2018 and March 2021, mostly from NYULH through EPIC and the MyChart patient portal. Additionally, recruitment methods included ClinicalTrials.gov and iConnect. Participants were considered eligible for our study if they were (1) English speaking; (2) 18–80 years old; (3) had a body mass index (BMI) between 27 and 50 kg/m^2^; (4) had prediabetes or moderately controlled T2D (5.7% [39 mmol/mol] ≤ hemoglobin A1c (HbA1C) ≤ 8% [64 mmol/mol]) managed with lifestyle alone or lifestyle plus metformin; and (5) had an estimated glomerular filtration rate (eGFR) > 60 mL/min/1.73 m^2^. Participants were excluded if they were non-English speaking; were unable or unwilling to participate in the study or accept their randomization group; became pregnant or planned to become pregnant during the study; had prior bariatric surgery or were unwilling to delay planned surgery for 12 months; were diagnosed with heart disease, kidney disease, or a chronic gastrointestinal disorder (e.g., inflammatory bowel disease); or were managing glycemia with insulin or a GLP-1 agonist (e.g., semaglutide). More detailed eligibility criteria are published elsewhere [8].

### 2.3. Screening, Data Collection Procedures, Randomization

Telephone pre-screening was conducted to establish eligibility and interest in the study. Prior to the COVID-19 pandemic, potentially eligible participants were scheduled for an in-person visit at the NYULH’s Clinical and Translational Science Institute to obtain informed consent and perform screening measurements (HbA1c, eGFR). In order to confirm BMI status, height was measured to the nearest 1 cm using a SECA 213 Stadiometer (Seca GmBH & Co. KG, Hamburg, Germany), and body weight was measured to the nearest 0.1 kg using a Stow weight scale (Scale-Tronix, Welch Allyn, Skaneateles, NY, USA). After March 2020, remote data collection methods were employed during the COVID-19 pandemic using WebEx videoconferencing (Cisco Webex, San Jose, CA, USA). Weight was measured using an electronic scale shipped to participants (RENPHO, Joicom Corporation, Eastvale, CA, USA), and data were transferred via Bluetooth to a corresponding mobile application, which study staff could access remotely. Height was self-reported. HbA1c and eGFR values were obtained from electronic medical records if performed within the last 6 months. Potential participants were instructed using WebEx videoconferencing to obtain point-of-care HbA1c with a provided A1cNow monitor (PTS Diagnostics, Whitestown, IN, USA). Kidney disease diagnosis was used in lieu of eGFR data if eGFR data were not available. We previously reported a subgroup analysis that included comparisons of weight loss within participants enrolled before and after the COVID-19 pandemic (March 2020) and found no difference in weight loss between the two groups [3].

Participants were randomized in blocks of four with equal allocation to one of two arms: *Standardized* or *Personalized*. Each study ID was preassigned to a randomized arm. Randomization was performed prior to the baseline assessment to allow for the timely collection and processing of the blood and stool samples required for the algorithm used to guide the *Personalized* diet. To ensure blinding and reduce bias, the study biostatistician performed randomization independent of measurements; data reported in this secondary analysis were derived from objective measures, and all participants remained blinded to the randomized arm until the fifth week of the study.

### 2.4. Baseline Measurements

Baseline data on age, sex, race/ethnicity, income, education, and medications were collected during the screening visit. Height and weight were also measured as described above, and these measurements were used to calculate the BMI. Weight-loss self-efficacy was measured using the validated, 20-item Weight Efficacy Lifestyle Questionnaire (WEL) [10]. Each item on the WEL includes a 10-point Visual Numeric Scale, with participants asked to rate their self-efficacy from 0 (not confident) to 9 (very confident). WEL items assess self-efficacy for resisting eating under various circumstances, including negative emotions, availability, social pressure, physical discomfort, and engaging in positive activities. An overall score and subscale scores were computed by summing the relevant questionnaire items. The overall WEL score was used in the analysis. Glycemic variability (GV) was measured for up to 14 days with a continuous glucose monitor (CGM; Abbott Freestyle Libre Pro, Abbott Park, IL, USA). We used the percentage of time above range (>140 mg/dL or TAR_>140_) to represent GV, as this value has been identified as an important metric for diabetes management in individuals with prediabetes or T2D not on insulin [11,12]

### 2.5. Interventions: Standardized vs. Personalized

Both Arms. Participants in both groups received an intensive behavioral counseling intervention program aimed at achieving a 7% weight loss. They were instructed to follow a calorie-restricted diet (specific details described below) and to engage in 150 min per week of moderate-to-vigorous physical activity while self-monitoring their diets using the Personal Nutrition Program (PNP) mobile application [13]. The PNP app provided real-time feedback regarding dietary intake in relation to the intervention targets specific to the participant’s randomization assignment (see below). Participants received individual PNP app training at the screening and baseline visits on how to use the app (e.g., enter foods, save meals), and staff were present throughout the study to assist participants with questions and troubleshooting. Participants engaged in registered dietitian-led, group-based, behavioral counseling, which was informed by Social Cognitive Theory and the Diabetes Prevention Program (DPP). Counseling was conducted via videoconferencing (WebEx) on a weekly basis for 4 weeks, then every other week for 20 weeks. Sessions were anchored by two brief (~5 min) animated videos, which were interspersed with open-ended questions designed to elicit discussion about experiences, values, and feelings; evaluate learning; and provide participants with the opportunity to practice applying new content. Additionally, the sessions included both educational content and behavioral strategies, as previously described [3,8].

Standardized. *Standardized* participants were advised to consume <25% of their calories from total dietary fat, which was adapted from the early Diabetes Prevention Program (DPP) and is at the lower end of the 2015–2020 Dietary Guidelines for Americans [14]. The PNP mobile app provided real-time feedback regarding calorie intake and the macronutrient distribution of meals and snacks logged by participants.

Personalized. *Personalized* participants received the same PNP mobile app feedback as the *Standardized* participants plus meal scores reflecting their predicted PPGR derived from a predictive algorithm. To generate meal scores, a battery of measures was collected prior to randomization, which included a stool sample (OMNIgene; OMR-200; DNA Genotek Inc., Ottawa, ON, Canada), HbA1c, anthropometric measurements, and sociodemographic- and health-related variables. Stool samples were shipped to the Weizmann Institute for microbiome analysis. Details regarding meal scores and the predictive algorithm have previously been published [3,8,13].

### 2.6. Statistical Analysis

We conducted descriptive analyses to compare baseline characteristics across intervention groups, presenting categorical variables as counts (proportions) and continuous variables as means (standard deviations). The predictive outcome of weight loss was defined as successful or a failure based on whether the weight loss at 6 months was ≥5% or <5%, as per the American College of Cardiology/American Heart Association Task Force on Practice Guidelines and The Obesity Society [13]. A weight loss of 5% is considered clinically meaningful and is associated with a significant reduction in metabolic risk associated with adiposity-based chronic diseases [15]. The predictors included baseline variables such as age, sex, race, ethnicity, income, education, BMI, metformin use, self-efficacy, and TAR_>140_. Missing values were imputed using random forest based on baseline variables, implemented via the *missForest* R package version 4.2.3. The gradient boosting machine method was used to train the predictive model with a five-fold cross-validation approach. The training set was used for model training and estimation, while the test set was used for evaluation. This procedure was repeated 100 times to ensure robustness. Evaluation metrics, including the area under the curve (AUC), sensitivity, and specificity, were calculated. This analysis was performed by combining the data from both arms (*Standardized* and *Personalized*) and separately within each study arm. The top 10 most important variables are reported; however, we will focus on the top 5 baseline characteristics. All statistical analyses were performed using R (R Core Team, R Foundation for Statistical Computing) version 4.2.3.

## 3. Results

This secondary analysis was based on 155 participants (*Personalized*: n = 84 vs. *Standardized*: n = 71) with complete baseline data (mean [standard deviation]: age, 59 [10] years; 66.5% female; 56.1% White; 23.1% African American, BMI 33.4 [4.6] kg/m^2^; HbA1c: 5.8 [0.59] %). There were no significant differences in baseline characteristics (Table 1), except that those randomized in the *Standardized* arm had a slightly lower BMI (Standardized BMI: 32.6 [4.22] kg/m^2^ vs. Personalized BMI: 34.2 [4.82] kg/m^2^, *p* = 0.03). Additionally, there was a difference in the distribution of sexes between the *Standardized* and *Personalized* arms that was approaching significance. Figure 1 shows the flow of participants.

### 3.1. Predictors in the Overall Sample

The predictive analysis using all 155 participants achieved an AUC of 0.68, a sensitivity of 0.79, and a specificity of 0.45. In the combined sample, 39.3% of participants successfully achieved clinically significant weight loss of ≥5% at 6 months. The top 10 most important baseline predictors for the combined sample are presented in Appendix A. Notably, weight-loss self-efficacy and age are the two most important variables in the predictive model. In the variable ranking importance, the accuracy of the model decreases by 21.4% if WEL is removed and 12.1% if age is removed from the model.

Comparing individuals who achieved successful weight loss at 6 months to those who did not (i.e., failure), of the five baseline characteristics, all were significantly different except for BMI (*p* = 0.08, Figure 2A–E).

### 3.2. Predictors in the Standardized Arm

The predictive models in the *Standardized* and Personalized arms achieved AUC of 0.58 and 0.62, sensitivity of 0.73 and 0.75, and specificity of 0.40 and 0.42, respectively. In the *Standardized* and *Personalized* arms, 39.4% and 39.8% were successful at achieving ≥5% weight loss at 6 months, respectively. The top 10 most important baseline predictors for the *Standardized* and Personalized arms are presented in Appendix A. Of the top five baseline characteristics in the *Standardized* arm, participants who achieved successful weight loss at 6 months had a higher baseline BMI (*p* < 0.0001; Figure 3A) and higher weight-loss self-efficacy (*p* = 0.02; Figure 3C) than those who did not achieve successful weight loss (i.e., failure). While age, TAR_>140_, and metformin use were top predictors in the predictive model, there was no significant difference in these variables between individuals who were successful with weight loss and those who were not successful (Figure 3B,D,E).

### 3.3. Predictors in the Personalized Arm

Those in the *Personalized* arm who achieved successful weight loss at 6 months had higher weight-loss self-efficacy (*p* < 0.0001; Figure 4A) and were older (*p* < 0.0001; Figure 4C) than those who did not achieve successful weight loss. However, in contrast to the *Standardized* arm, BMI was not significantly different between those who achieved successful weight loss and those who did not (i.e., failure; Figure 4B) in those randomized to the *Personalized* arm. TAR_>140_ and sex were both top five predictors; however, neither was statistically significant for weight-loss failure or weight-loss success (Figure 4D,E).

## 4. Discussion

The purpose of this secondary analysis was to identify baseline characteristics that predict weight-loss success (≥5% initial body weight) in adults with prediabetes and obesity. We found similar baseline predictors within each arm, including age, weight-loss self-efficacy, GV measured as TAR_>140_, metformin use, and BMI. However, the main findings from the analysis showed that only baseline weight-loss self-efficacy, measured through the WEL questionnaire, was a predictor of weight-loss success in both study arms.

Weight-loss self-efficacy is the ability to control eating in challenging situations and is a strong predictor and mediator of weight management [16]. In our analysis, the contribution, or level of importance, of weight-loss self-efficacy to the predictive model was greater in the *Personalized* arm than in the *Standardized* arm. More specifically, the accuracy of the model decreases by 19.4% in the *Personalized* arm if weight-loss self-efficacy is removed from the predictive model, compared to a 9.27% decrease in the *Standardized* arm. Our results support findings that greater self-efficacy is associated with greater weight loss [16,17]. In particular, Shin et al. (2011) dichotomized early postmenopausal women into successful and unsuccessful weight loss at 6 months [18]. They found that those who demonstrated higher self-efficacy, measured through the WEL questionnaire, had a greater likelihood of >5% weight loss at 6 months. In adults with T2D, those randomized to a behavioral weight-loss intervention, similar to our study, showed greater improvements in weight-loss self-efficacy as measured by the WEL questionnaire, which corresponded with greater weight loss and improvement in HbA1c levels [19]. Low weight-loss self-efficacy is associated with unfavorable eating behaviors in adults with overweight and obesity. Oikarinen et al. (2023) examined eating self-efficacy in adults with overweight and obesity and found that those with low WEL scores had lower cognitive restraint and higher uncontrolled eating, emotional eating, and binge eating than participants classified with high WEL scores [20]. We should note an imbalance in the proportion of females and males in the *Personalized* arm, which may have impacted our findings, given that there are differences in self-efficacy between females and males. Presenell et al. (2008) found that higher baseline self-efficacy predicted greater weight loss in men but not in women [17]. Females are more likely to report a history of frequent dieting, a proxy for repeated diet failures (i.e., weight cycling), which may undermine females’ confidence in maintaining healthy eating and body weight successfully [21].

While our data support the role of baseline self-efficacy on weight loss, it is equally important to enhance self-confidence in one’s ability to control eating for those with lower self-efficacy. In a 12-month, randomized trial in adults with overweight or obesity, Nezami et al. applied Social Cognitive Theory techniques, including goal setting, self-monitoring, identifying barriers, and problem-solving, to increase self-efficacy in eating and physical activity [22]. They found that greater increases in eating self-efficacy (e.g., WEL scores) were associated with reduced caloric intake and greater weight loss at 12 months. These findings are supported by others [23,24]. Future weight-loss interventions and predictive algorithms should incorporate weight-loss self-efficacy as a predictor of weight-loss success. Additionally, researchers should include theory-based behavioral approaches, such as Social Cognitive Theory, tailored to weight-loss self-efficacy at baseline or consider stratifying by self-efficacy score.

We identified unique patterns specific to each arm through our secondary analysis, which may inform future weight-loss interventions. In the *Standardized* arm, individuals with a higher BMI (~35 kg/m^2^) achieved greater weight-loss success than those with a lower BMI (~30 kg/m^2^). BMI is a collectively poor measure of metabolic health and adiposity, but it offers a glimpse into the potential impact on screening and delivering an effective intervention. In contrast, those in the *Personalized* arm who were older achieved successful weight loss at 6 months. Older age is a predictor of weight-loss success in adults included in randomized clinical trials, which may be explained by less family responsibility or fewer work demands, resulting in greater adherence to the intervention requirements [25]. Mechanistically, it is unclear why older adults lose more body weight in behavioral weight-loss interventions. This is not explained by changes in the resting metabolic rate, which begins to decline after age 60 [26]. One hypothesis is that peak fat mass occurs after the fifth decade of life, with female fat mass peaking at 50 years and male fat mass peaking at 75 years [27]. As such, older adults may have more fat mass (e.g., excess adipose tissue) to lose. In the current study, we speculate that older adults may fare better in the *Personalized* arm, as they have more time and fewer life responsibilities (i.e., work, children), allowing them to dedicate more time to self-monitoring, a key aspect of the intervention.

Designing precision nutrition studies may require subgroup classifications for a more tailored and efficacious intervention. Precision nutrition interventions may not be about developing tailored diet recommendations for individuals but rather about stratifying individuals based on specific biomarkers and metabolic factors to better estimate and provide dietary recommendations [28]. For example, Wagner et al. identified six metabolic subgroups, or *metabotypes,* through extensive phenotyping using intra-individual responses to a battery of variables, such as an oral glucose tolerance test and magnetic resonance imaging-measured body fat, with varying risks associated with the development of T2D, kidney disease, and all-cause mortality [29]. Hillesheim et al. delivered a 12-week parallel-arm, randomized controlled trial in healthy adults, classifying them into *metabotypes* using four biomarkers (triacylglycerol, high-density lipoprotein-cholesterol, total cholesterol, and glucose) [30]. In this study, participants randomized to the personalized group received dietary advice based on a clustering model derived from the four aforementioned biomarkers. Despite improvements in diet quality and other metabolic markers (triacylglycerol, total cholesterol), weight was not significantly different between the personalized group (i.e., sub-grouped into metabotypes) and the control group, which received generic dietary advice based on national guidelines.

Tailoring precision nutrition predictive algorithms for obesity treatment likely requires the inclusion of features that are components of the energy balance model, such as total energy expenditure, fat mass, and appetitive hormones (e.g., glucagon-like peptide-1). Acosta et al. categorized over 450 participants with obesity into four phenotypes: hungry brain, emotional hunger, hungry gut, and slow burn [31]. In a pragmatic clinical trial, the phenotype-guided approach was associated with a 1.75-fold greater weight loss after 12 months. These findings, like those from our secondary analysis, underscore the importance of integrating both biological and behavioral data in future obesity interventions.

The strengths of our secondary analysis include randomization, testing an innovative precision nutrition intervention, and long-term follow-up at 6 months in a free-living sample of middle-aged to older adults with prediabetes and obesity. Additionally, the sensitivity of the predictive models used in this analysis was high (>0.7) in all situations, indicating a strong ability to identify participants who successfully lost weight correctly. Despite our strengths, our study had several limitations. While our prior subgroup analyses indicate no effect of the COVID-19 pandemic on weight loss, self-quarantine measures may have affected lifestyle behaviors [32]. Additionally, the use of remote measurements may have inadvertently introduced bias. The specificity of the predictive models was low (< 0.5), which could be due to the unbalanced proportions (weight-loss success < weight-loss failure). From a clinical standpoint, our findings may serve as a screening tool to help identify patients who may be successful with weight loss, although, given the low specificity in our model, healthcare providers should consider other characteristics of the patient that may prevent weight-loss success (e.g., social determinants of health). We used a binary 5% cutoff, which is clinically relevant; however, metabolic benefits have been documented with weight losses of 3% [33]. We are limited by the predictors included in the analysis, and there may have been measurements that we did not include that could be relevant to weight-loss success. For example, in a systematic review, Chopra et al. found that limiting dietary fat intake was significantly associated with successful weight loss in individuals undergoing lifestyle interventions [25]. Our sample mostly consisted of White females who made >$75,000 per year, with <20% of our sample self-reporting as Hispanic. As a result, this limits the generalizability of our findings to other populations, particular those of low socioeconomic status. Additionally, there were incomplete data for the GV measures, resulting in 20% missing data from our sample, which may have impacted our results.

## 5. Conclusions

In conclusion, our secondary analysis provides valuable insights into understanding the baseline characteristics of adults with prediabetes and obesity who are included in precision nutrition interventions. Our findings may help researchers identify predictors of weight-loss success that can be used to stratify large samples into smaller, unique subgroups, providing early support and tailoring and promoting more efficacious results. We propose that algorithms used in precision nutrition interventions should incorporate both behavioral and biological data into their predictive models and consider including age, BMI, and weight-loss self-efficacy.

## Figures and Tables

**Figure 1 nutrients-17-02178-f001:**
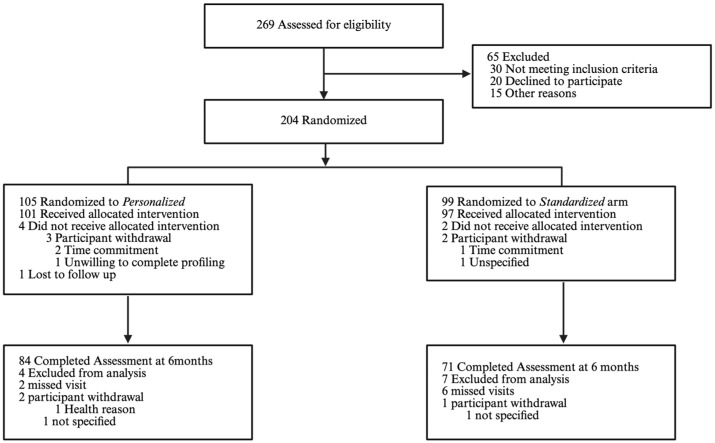
Participant enrollment flow.

**Figure 2 nutrients-17-02178-f002:**
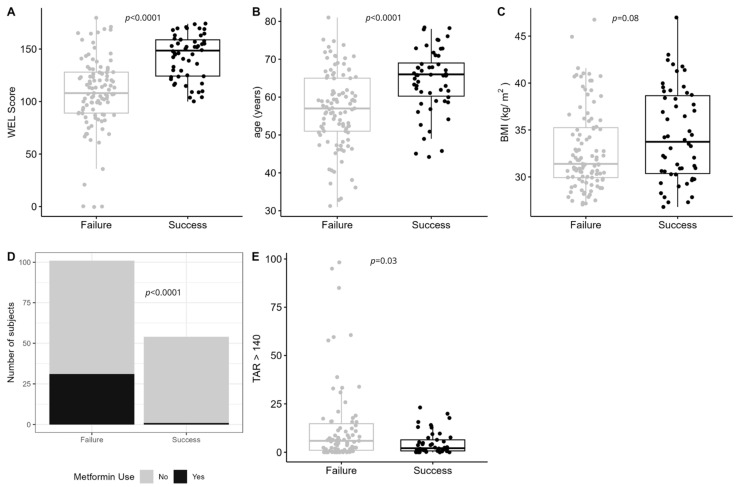
The top five baseline predictors of weight-loss success in the combined sample. Success was defined as those who achieved ≥5% weight loss at 6 months, whereas failure was defined as those with <5% weight loss. (**A**) Weight-loss self-efficacy (WEL) score (*p* < 0.0001); (**B**) age at study visit in years (*p* < 0.0001); (**C**) body mass index (BMI, *p* = 0.08); (**D**) number of participants on metformin at randomization (*p* < 0.0001); and (**E**) glycemic variability measured as time above range (>140 mg/dL, TAR_>140_ (*p* = 0.03).

**Figure 3 nutrients-17-02178-f003:**
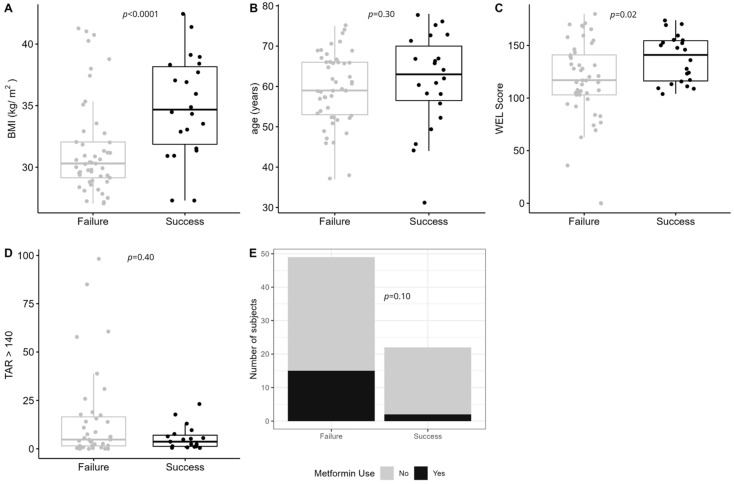
The top five baseline predictors of weight-loss success in the Standardized arm. Success was defined as those who achieved ≥5% weight loss at 6 months, whereas failure was defined as those with <5% weight loss. (**A**) Body mass index (BMI, *p* < 0.0001); (**B**) age at study visit in years (*p* = 0.3); (**C**) weight-loss self-efficacy (WEL) score (*p* = 0.02); (**D**) glycemic variability measured as time above range (>140 mg/dL, TAR_>140_ (*p* = 0.4); and (**E**) number of participants on metformin at randomization (*p* = 0.10).

**Figure 4 nutrients-17-02178-f004:**
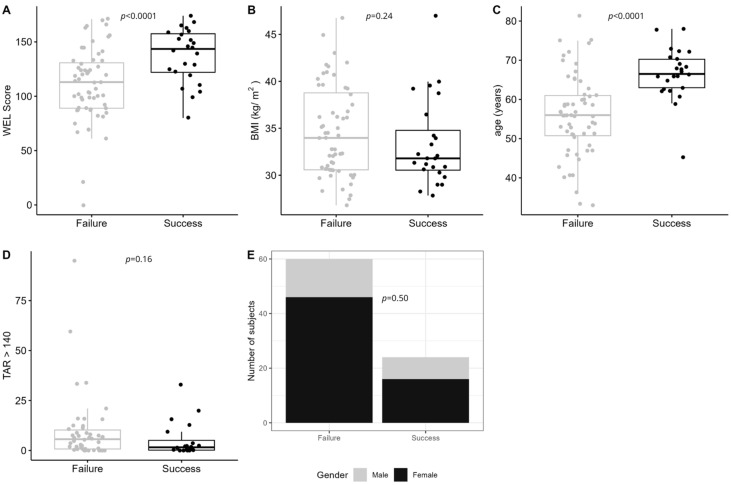
The top five baseline predictors of weight-loss success in the Standardized arm. Success was defined as those who achieved ≥5% weight loss at 6 months, whereas failure was defined as those with <5% weight loss. (**A**) Weight-loss self-efficacy (WEL) score (*p* < 0.0001); (**B**) body mass index (BMI, *p* = 0.24); (**C**) age at study visit in years (*p* < 0.0001); (**D**) glycemic variability measured as time above range (>140 mg/dL, TAR_>140_ (*p* = 0.16); and (**E**) female sex (*p* = 0.5).

**Table 1 nutrients-17-02178-t001:** Baseline demographics, anthropometrics, and metabolic variables.

	Mean (SD)			
Characteristic	All (n = 155)	Standardized (n = 71)	Personalized (n = 84)	*p*-Value
Sex, No. (%)				0.05
Female	103 (66.5%)	41 (57.7%)	62 (73.8%)	
Male	52 (33.5%)	30 (42.3%)	22 (26.2%)	
Age, y	59.5 (10.2)	60.0 (9.87)	59.0 (10.6)	0.55
Race/ethnicity, No. (%)				0.40
White	87 (56.1%)	44 (62.0%)	43 (51.2%)	
African American	36 (23.2%)	16 (22.5%)	20 (23.8%)	
Other ^‡^	30 (19.4%)	11 (15.5%)	19 (22.6%)	
Missing	2 (1.3%)	0 (0%)	2 (2.4%)	
Ethnicity				1.0
Non-Hispanic	130 (83.9%)	60 (84.5%)	70 (83.3%)	
Hispanic	25 (16.1%)	11 (15.5%)	14 (16.7%)	
Education, No (%)				0.75
<Bachelor’s Degree	40 (25.8%)	20 (28.2%)	20 (23.8%)	
≥Bachelor’s Degree	106 (68.4%)	48 (67.6%)	58 (69.0%)	
Missing	9 (5.8%)	3 (4.2%)	6 (7.1%)	
Income, per year, No (%)				0.93
<$75,000	52 (33.5%)	25 (35.2%)	27 (32.1%)	
>$75,000	81 (52.3%)	37 (52.1%)	44 (52.4%)	
Missing	22 (14.2%)	9 (12.7%)	13 (15.5%)	
Body weight, kg	92.8 (16.4)	91.1 (15.5)	94.2 (17.2)	0.24
BMI, kg/m^2^	33.4 (4.61)	32.6 (4.22)	34.2 (4.82	0.03
Weight-loss self-efficacy, total score	120 (34.7)	123 (36.3)	118 (33.4)	0.44
TAR_>140_, %	10.0 (17.4)	11.8 (19.8)	8.41 (15.0)	0.29
HbA1c, %	5.80 (0.588)	5.84 (0.603)	5.78 (0.578)	0.53
Metformin use, No. (%)	32 (20.6%)	17 (23.9%)	15 (17.9%)	0.46

TAR_>140_: Time above range is the percentage of time above 140 mg/dL; TAR_>140_: n = 31 (20%) missing overall; Standardized: n = 13 (18.3%); Personalized: n = 18 (21.4%); ^‡^ Other: includes Asian, Native Hawaiian or other, Pacific Islander, unknown race or ethnicity, and race or ethnicity not reported; *p* < 0.05.

## Data Availability

De-identified data will be made available upon request to the corresponding author (collin.popp@nyulangone.org).

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
