# Peer review of "Baseline Characteristics of Weight-Loss Success in a Personalized Nutrition Intervention: A Secondary Analysis"

_nutrients, 2025, doi:10.3390/nu17132178_

Round 1

Reviewer 1 Report

Comments and Suggestions for Authors

This paper conducts a secondary analysis of a personalized nutrition intervention trial to explore the relationship between baseline characteristics and weight loss success. However, the methodological aspects of the paper still require further refinement. Below are the specific comments: 

The research objective is clear, aiming to identify baseline characteristics associated with successful weight loss using machine learning methods. However, the introduction could further emphasize the controversial findings in current research on personalized nutrition interventions to highlight the novelty and necessity of this study. 

The methods section provides a detailed description of the study design, participant recruitment, randomization process, and intervention measures, demonstrating high transparency and reproducibility. Nevertheless, the remote data collection methods (e.g., weight and height measurements during the pandemic) may introduce bias, but the paper does not sufficiently discuss the potential impact of such bias on the results. Although the handling of missing data (e.g., missing glycemic variability data) is mentioned, the specific imputation methods or sensitivity analyses are not described. It is recommended to supplement this information to enhance the robustness of the findings. In lines 99–100, the authors could add a reference to support the statement: "Referring to previous literature [doi: 10.3390/nu16060777], baseline data on age, sex, race/ethnicity, income, education, and medications were collected during the screening visit." 

The results section presents baseline characteristics, predictive model performance metrics, and differences in characteristics among successful weight loss participants across intervention groups. The figures and tables are clear. However, although the model shows high sensitivity, its specificity is relatively low. The authors should further discuss the potential implications of this imbalance for clinical applications. Additionally, the stronger association between age and weight loss success in the personalized intervention group is not explored in terms of underlying mechanisms (e.g., adherence or metabolic rate changes), which should be addressed in the discussion. 

The discussion summarizes the key findings and compares them with existing literature, particularly regarding the role of weight loss self-efficacy. However, the authors note that older participants in the personalized intervention group were more likely to achieve weight loss success but do not sufficiently discuss possible reasons (e.g., differences in social roles or time commitment). While the importance of weight loss self-efficacy is emphasized, no specific intervention recommendations (e.g., how to enhance self-efficacy) are provided. The limitations section could more comprehensively address the impact of limited sample diversity (e.g., uneven distribution of race and gender) on the generalizability of the results. 

The variable importance ranking table in the supplementary materials should be described in greater detail, including the specific contribution or weight of each variable. The statement "future interventions should consider providing support based on baseline characteristics" does not specify the form of support (e.g., psychological intervention or personalized feedback). It is recommended to clarify this. Some abbreviations (e.g., TAR>140) should be spelled out upon first use for better readability.

Author Response

We sincerely thank the reviewer for their time and dedication to reviewing our paper. We recognize this review is done on their own time and in-kind; we appreciate their contribution to science, especially in the area of precision nutrition.

Reviewer comment: The research objective is clear, aiming to identify baseline characteristics associated with successful weight loss using machine learning methods. However, the introduction could further emphasize the controversial findings in current research on personalized nutrition interventions to highlight the novelty and necessity of this study.

Author response: We thank the reviewer for the comment and have added a discussion on the controversial findings of the current literature. See lines 45-60.

Reviewer comment: The methods section provides a detailed description of the study design, participant recruitment, randomization process, and intervention measures, demonstrating high transparency and reproducibility. Nevertheless, the remote data collection methods (e.g., weight and height measurements during the pandemic) may introduce bias, Still, the paper does not sufficiently discuss the potential impact of such bias on the results.

Author response: We thank the reviewer for the comment regarding potential bias in the report data collection during COVID-19. We add clarification in lines 109-112. Additionally, we will add COVID-19 as a limitation and the potential introduction of bias into our results. See lines 347-350.

Reviewer comment: Although the handling of missing data (e.g., missing glycemic variability data) is mentioned, the specific imputation methods or sensitivity analyses are not described. It is recommended to supplement this information to enhance the robustness of the findings.

Author response: We thank the reviewer for the comment. To address missing values, we used random forest imputation, which is well-suited for handling mixed-type data, including both categorical and continuous variables. This imputation was performed using the missForest R package. We have added this information to the Statistical Analysis

Reviewer comment: In lines 99–100, the authors could add a reference to support the statement: "Referring to previous literature [doi: 10.3390/nu16060777], baseline data on age, sex, race/ethnicity, income, education, and medications were collected during the screening visit." 

Author response: We thank the reviewer for the suggestion, but do not find it necessary to add a reference to support our statement on previous literature, as the predictors included were based on available baseline measures.

Reviewer comment: The results section presents baseline characteristics, predictive model performance metrics, and differences in characteristics among successful weight loss participants across intervention groups. The figures and tables are clear. However, although the model shows high sensitivity, its specificity is relatively low. The authors should further discuss the potential implications of this imbalance for clinical applications.

Author response: We acknowledged the low specificity of the predictive model in the limitations section (please see lines 350-351), which indicates that the model may predict increased false positives (may predict that individuals will have weight loss success when they in fact will have weight loss failure). Additionally, we added a few points on the clinical relevance in lines 352-355. The predictive model was developed to identify characteristics prior to a weight loss intervention that may inform early intervention strategies for personalized support. While this model provides evidence of characteristics that may predict weight loss success, we would encourage clinicians to engage in weight loss monitoring to promote the most efficacious results. In addition, other participant characteristics both at baseline and throughout the intervention are important to determine weight loss success.

Reviewer comment: Additionally, the stronger association between age and weight loss success in the personalized intervention group is not explored in terms of underlying mechanisms (e.g., adherence or metabolic rate changes), which should be addressed in the discussion.

Author response: We thank the reviewer for this thoughtful comment, and we do discuss older adults adhering more to a personalized diet due to availability and more time. We added to our discussion a few points regarding potential mechanisms associated with age and weight loss success. See lines 309-313.

Reviewer comment: The discussion summarizes the key findings and compares them with existing literature, particularly regarding the role of weight loss self-efficacy. However, the authors note that older participants in the personalized intervention group were more likely to achieve weight loss success but do not sufficiently discuss possible reasons (e.g., differences in social roles or time commitment).

Author response: We thank the reviewer for pointing this out. We discuss this at the end of the 5th paragraph in the discussion section (Lines 314-316).

Reviewer comment: While the importance of weight loss self-efficacy is emphasized, no specific intervention recommendations (e.g., how to enhance self-efficacy) are provided.

Author response: We thank the reviewer for their comment. While we are not looking at changes in self-efficacy, we are showing that a high level of self-efficacy at baseline results in greater weight loss success. However, we added additional information on ways to increase self-efficacy. See lines 288-299.

Reviewer comment: The limitations section could more comprehensively address the impact of limited sample diversity (e.g., uneven distribution of race and gender) on the generalizability of the results.

Author response: We thank the reviewer for their comment and added a point regarding generalizability to the discussion section (Lines 361-364)

Reviewer comment: The variable importance ranking table in the supplementary materials should be described in greater detail, including the specific contribution or weight of each variable.

Author response: We thank the reviewer for the comment and have added a description of the variable importance ranking in the supplemental material.

Reviewer comment: The statement "future interventions should consider providing support based on baseline characteristics" does not specify the form of support (e.g., psychological intervention or personalized feedback). It is recommended to clarify this.

Author response: Thank you for the comment. We clarified the issue by adding “behavioral support.”

Reviewer comment: Some abbreviations (e.g., TAR>140) should be spelled out upon first use for better readability.

Author response: TAR is first mentioned in line 134, and is spelled out. We reviewed the manuscript to ensure all abbreviations are spelled out upon first use.

Reviewer 2 Report

Comments and Suggestions for Authors

Thank you for the opportunity to review this manuscript. This is well-written. Interesting research for readers. I have only minor comments.

“interventions on weight loss in participants with prediabetes or moderately controlled type 2 diabetes (T2D)”
→ Consider clarifying what is meant by “moderately controlled” T2D. Is there a specific HbA1c range or clinical criteria?

“More detailed eligibility criteria were published elsewhere [6].”
→ Consider briefly summarizing any major exclusion criteria here (e.g., insulin use, recent weight loss surgery), if space allows.

“Remote data collection methods were employed during the COVID-19 pandemic…”

You might clarify the time period during which remote collection was implemented,

“self-monitoring their diets using the Personal Nutrition Program (PNP) mobile application…”
→ Suggest specifying whether participants received training or support for using the app, especially among older or less tech-savvy individuals.

“Standardized participants were advised to consume <25% of their calories from total dietary fat.”
→ You could clarify whether this recommendation aligns with a specific dietary guideline

“weight loss at 6 months was ≥5% or <5%... as per ACC/AHA/Obesity Society…”
→ A citation is given, but it may be helpful to briefly state why 5% was chosen (e.g., clinically meaningful weight loss threshold)

Author Response

We sincerely thank the reviewer for their time and dedication to reviewing our paper. We recognize this review is done on their own time and in-kind; we appreciate their contribution to science, especially in the area of precision nutrition.

Reviewer Comment: “interventions on weight loss in participants with prediabetes or moderately controlled type 2 diabetes (T2D)”
→ Consider clarifying what is meant by “moderately controlled” Is there a specific HbA1c range or clinical criteria?

Author response: This is defined in lines 82-83: “moderately controlled type 2 diabetes (defined as a hemoglobin A1c level ≤8.0% while managed with lifestyle modification alone or lifestyle modification plus metformin)”

Reviewer Comment:“More detailed eligibility criteria were published elsewhere [6].”
→ Consider briefly summarizing any major exclusion criteria here (e.g., insulin use, recent weight loss surgery), if space allows.

Author response: We thank the reviewer for their comment. We provided additional details regarding the exclusion criteria, see lines 84-91.

Reviewer Comment: “Remote data collection methods were employed during the COVID-19 pandemic…” You might clarify the time period during which remote collection was implemented,

Author response: We added additional information regarding the start of our remote data collection methods (Line 100).

Reviewer Comment: “self-monitoring their diets using the Personal Nutrition Program (PNP) mobile application…”
→ Suggest specifying whether participants received training or support for using the app, especially among older or less tech-savvy individuals.

Author response: We thank the reviewer for their comment and added the following: “Participants received individual PNP app training at the screening and baseline visits on how to use the app (e.g., enter foods, save meals), and staff were present throughout the study to assist participants with questions and troubleshooting.” See lines 144-146.

Reviewer Comment: “Standardized participants were advised to consume <25% of their calories from total dietary fat.”
→ You could clarify whether this recommendation aligns with a specific dietary guideline

Author response: We thank the reviewer for the comment and added additional information on where the 25% kcals from fat were derived. See lines 156-157.

Reviewer Comment: “weight loss at 6 months was ≥5% or <5%... as per ACC/AHA/Obesity Society…”
→ A citation is given, but it may be helpful to briefly state why 5% was chosen (e.g., clinically meaningful weight loss threshold).

Author response: We thank the reviewer for the comment and added the following: “Weight loss of 5% is considered clinically meaningful and is associated with a significant reduction in metabolic risk associated with adiposity-based chronic diseases [Magkos F, et al 20216].” See lines 175-176.